



# Inter-calibrating SMMR brightness temperatures over continental surfaces.

Samuel Favrichon[1,2], Carlos Jimenez[2,1], and Catherine Prigent[1,2]

[1]Sorbonne Université, Observatoire de Paris, Université PSL, CNRS, LERMA, Paris, France
[2]Estellus, Paris, France

**Correspondence:** Favrichon Samuel (samuel.favrichon@obspm.fr)

**Abstract.**

Microwave remote sensing can be used to monitor the time evolution of some key parameters over land, such as land surface temperature or surface water extent. Observations are made with instrument such as the Scanning Microwave Multichannel Radiometer (SMMR) before 1987, the Special Sensor Microwave / Imager (SSM/I) and the following Special Sensor Microwave Imager / Sounder (SSMIS) from 1987 and still operating, to the more recent Global Precipitation Mission Microwave Imager (GMI). As these instruments differ on some of their characteristics and use different calibration schemes, they need to be inter-calibrated before long time series products can be derived from the observations. Here an inter-calibration method is designed to remove major inconsistencies between the SMMR and other microwave radiometers for the 18GHz and 37 GHz channels over continental surfaces. Because of a small overlap in observations and a ∼6 h difference in overpassing times between SMMR and SSM/I, GMI was chosen as a reference despite the lack of a common observing period. The diurnal cycles from three years of GMI brightness temperatures are first calculated, and then used to evaluate SMMR differences. Based on a statistical analysis of the differences, a simple linear correction is implemented to calibrate SMMR on GMI. This correction is shown to also reduce the biases between SMMR and SSM/I, and can then be applied to SMMR observations to make them more coherent with existing data record of microwave brightness temperatures over continental surfaces.

## 1 Introduction

Since 1978, passive microwave satellite imagers provide Earth observations at multiple frequencies, over ocean and land, for atmospheric or surface applications such as cloud and precipitation monitoring, surface temperature estimation, ocean wind speed or sea ice concentration retrievals (Ulaby et al. 1986). With now more than 40 years of data record, climate analysis can be performed from these measurements, provided that the observation time series are well calibrated, consistent, and homogeneous.





The successive microwave imagers share common characteristics, but with technological changes from a generation to the next and possible calibration issues between instruments even from the same series. Major microwave imagers include the Seasat Scanning Multichannel Microwave Radiometer (SMMR) from 1978 to 1987 (Gloersen and Barath 1977), the Special Sensor Microwave Imagers (SSM/I) and the Special Sensor Microwave Imager / Sounder (SSMIS) from 1987 up to

now (Hollinger et al. 1990), the Advanced Microwave Scanning Radiometer (AMSR-E and AMSR2) since 2002 (Kawanishi et al. 2003), or the more recent Global Precipitation Microwave Imager (GMI) since 2014 (Hou et al. 2014). To create homogeneous measurements from different instruments, each instrument has to be carefully calibrated first and the different instruments have then to be inter-calibrated. The changes between instruments include differences in overpassing times, in Earth Incidence Angles (EIA), in channel center frequencies or in bandwidths. All these aspects need to be accounted for in

the inter-calibration process.

Multiple teams have worked to provide corrections for the brightness temperatures of microwave imagers to ensure homogeneous data records over time  (e.g., Berg et al. 2013; Wentz 2013; Fennig et al. 2019) than can be used as Fundamental Climate Data Records (FCDR). FCDR are calibrated data from multiple sensors that have been made coherent and quality controlled to be accurate and stable over long time periods. These are the basic data records from which geophysical products can be

computed and aggregated to form long term records of Essential Climate Variables (Bojinski et al. 2014).

Different methods exist to inter-calibrate sensors. Quasi-direct comparisons of observations can be performed at the poles, where overpassing of polar satellites are frequent and the surface responses rather stable. However, this calibration is limited to rather cold brightness temperatures (Sapiano et al. 2013). The statistics of the coldest or warmest scenes have also been analyzed to inter-calibrate the sensors. This is the so-called vicarious calibration, applied over ocean for the cold end (Ruf

2000) and on the Amazon forest for the warm end  (Brown and Ruf 2005).

More recently this vicarious method has been extended to other forested sites, taking into account their seasonal variability (Yang et al. 2016). Over ocean, double difference methods are often adopted, using radiative transfer simulations as the reference, to bridge the gap between instruments with different characteristics (Kroodsma et al. 2012). The radiative transfer model is usually fed by atmospheric and surface information from reanalyses and it can theoretically account for changes in

EIA, in channel characteristics (e.g., frequency, bandwidth), as well as differences in overpassing times. This method is very challenging over continental surfaces. First, microwave radiative transfer models over land, along with all their necessary input parameters (e.g., soil moisture, vegetation density, snow water equivalent), are not available with the required quality over a large range of surface types. Second, the possibly strong diurnal variability of the land surface temperatures is not described with enough accuracy and temporal resolution to account for differences in the satellite overpassing times. Nevertheless, Dai

and Che (2009) tested a modeling of the diurnal variation of the surface temperature to inter-calibrate instruments with different over-passing times over land.

Another inter-calibration method consists in using matchups with a reference instrument that has a different orbit type, making it possible to provide quasi-direct comparisons over a large range of latitudes, even for satellites with different overpassing times. With their low orbits that sample the diurnal cycle, the Tropical Rainfall Measuring Mission Microwave Imager (TMI),

and more recently the Global Precipitation Mission Microwave Instrument (GMI) can be used as intermediate references to





inter-calibrate Sun-synchronous instruments with drastically different overpassing times such as the SSM/I series (equator overpassing time around dusk and dawn) and the AMSR series (equator overpassing times at midnight and midday). Berg et al. (2018) applied the different methods to inter-calibrate the SSM/I series. The different methods agreed well with each other, offering an increased confidence in the proposed inter-calibration.

| Instrument | Reference | SMMR | SSM/I | SSMIS | TMI | AMSR-E | AMSR2 | GMI |
|---|---|---|---|---|---|---|---|---|
| CDR CSU V01 | Berg et al. (2013) | - | ✓ | ✓ | ✳ | - | - | - |
| CSU V03 | Berg et al. (2018) | - | ✓ | ✓ | ✓ | ✓ | ✓ | ✳ |
| CDR RSS | Wentz (2013) | - | ✓ | ✓ | - | - | - | - |
| CM SAF FCDR | Fennig et al. (2019) | ✓ | ✳ | ✓ | - | - | - | - |

**Table 1.** The characteristics of the available Fundamental Climate Data Records from passive microwave imagers. The ✓ denotes an inter-calibrated instrument while the ✳ is for the reference instrument.

Table 1 lists the major FCDRs from passive microwave imagers available to the community, indicating the inter-calibrated instruments, and the reference instrument. EUMETSAT Climate Monitoring Satellite Application Facility (CM SAF), Colorado State University (CSU), and Remote Sensing Systems (RSS) all include SSM/I and SSMIS in their inter-calibrations. CSU uses for the first time GMI in their recent inter-calibration scheme (Berg et al. 2018). They simultaneously employ multiple calibrations methods to reduce the uncertainty on the data record. Their initial work (V01) is improved by adding more satellites

and by using GMI as the reference for its description of the diurnal cycle over land. RSS inter-calibration effort is essentially based on the use of a radiative transfer model over ocean (Wentz 2013). The CM SAF uses SSM/I F11 as their reference satellite. The inter-calibration is a scene dependent correction. It is done by correcting the cold end of the observed $SMMR - ERA\ T_b$ differences to match the $SSM/I - ERA\ T_b$ differences but keeping the warm calibration end at the observed hot load target temperature. It is therefore not expected to have an impact on the warm $T_b$ range.

So far, only the CM SAF includes SMMR in their FCDR. The inter-calibration was developed initially for the monitoring of fluxes over the ice-free ocean (Andersson et al. 2010). To extend the climate record of satellite-derived land surface parameters in time, here we propose to analyze the possibility of inter-calibrating the SMMR instrument over land. The SMMR instrument failed in August 1987 and the first SSM/I (F08) was launched in 1987, with an overlapping time of only a few weeks, and with ∼6 h differences in their overpassing times at the equator. Here, we suggest to use GMI as a reference instrument, assuming

that the environmental conditions have not changed drastically from the SMMR to the GMI era, to allow the comparison of a



large set of observations averaged over time. This strategy does not allow to perform a detailed inter-calibration but it makes it possible to correct for major biases that so far hamper the use of SMMR over land for the generation of climate record of geophysical parameters. We will concentrate on the channels that are common to all the microwave imagers used in the FCDR in Table 1, the Ku (around 18 GHz) and Ka (around 36 GHz) channels. These are key observations for the retrieval of several

land surface parameters (e.g., surface water extent (Prigent et al. 2007), snow water equivalent (Pulliainen 2006), or land surface temperature (Jiménez et al. 2017)). We will use the CM SAF FCDR datasets as the starting point of our developments.

In Section 2, the satellite observations used in this study are briefly described, along with their preprocessing for the analysis. The result of the inter-comparison is presented in Section 3, along with the proposed inter-calibration procedure and its evaluation. Section 4 concludes this study.

## 10  2   Data and Method

### 2.1   The satellite data

Table 2 summarizes the major characteristics of the SMMR and GMI instruments for the Ku and Ka channels. SSM/I on board F08 is also included, as it is the only instrument with an overlap period with SMMR (albeit of only 28 days) and it will be used for evaluation. GMI, with its non Sun-synchronous orbit, observes the full diurnal cycle, including the SMMR and SSM/I

overpassing times.

SMMR was launched in 1978 on the Nimbus-7 satellite, and operated until August 1987. It is the first multichannel microwave imager, designed mainly for oceanic applications to estimate the surface wind speed and sea surface temperature (Gloersen and Barath 1977). Due to power limitation onboard, measurements were performed only every other day. The EIA decreases from 50.2°to 49.3°starting in 1986. The initial SMMR record used by the CM SAF is the Level 1B data, not the raw

counts as for the other instruments. The SMMR Level 1B data is described by Njoku et al. (1980) and it includes the antenna pattern and spill-over correction, as well as sensor drift correction. The calibration uses the cosmic background temperature as a cold reference (2.7 K), an on-board hot calibration load around 300 K as well as climatological means to estimate biases in the calibrations. A linear calibration was performed to reduce instrument bias in the target domain, i.e. for ocean surface parameter estimations. The inter-calibration performed by the CM SAF is based on the double difference technique between the

SSM/I F08 and SMMR brightness temperatures ($T_{bs}$), using radiative transfer simulations from reanalysis to account for the changes in frequencies, bandwidths, and EIA. This correction is computed only over cloud-free water surfaces. It is described in details in Fennig et al. (2019). In this study, we will only use SMMR data showing the best quality (data with sun intrusion, field-of-view, or scan error are removed).

GMI is a recent microwave imager launched in 2015. It observes between 10 and 190 GHz to measure precipitation across

the globe. It will be used here as a reference standard for calibration. The instrument characteristics are summarized in Table 2, listing only the channels relevant to this study. The satellite has a 65°inclination, allowing non Sun-synchronous observations of the Earth, from the tropics to the high latitudes. The antenna has an EIA of 52.8°and a swath of ~900 km. This swath width associated with the inclination means that polar regions are not fully covered. The GMI calibration is described by Wentz





| Instrument | SMMR | SSM/I (F08) | GMI |
|---|---|---|---|
| Earth Incidence Angle (°) | 50.2 to 49.3* | 53.1 | 52.8 |
| Channels (GHz) | 18.0 (V,H), 21.0 (V,H)*, 37 (V,H) | 19.35 (V,H), 22.235 (V), 37 (V,H) | 18.7 (V,H), 23.8 (V), 36.5 (V,H) |
| Instantaneous Field Of View (km x km at 37 GHz) | 17×29 | 24×36 | 8.6×14 |
| Ascending equator overpassing time (h) | 00:00 | 06:00 | non Sun-synchronous |
| Operating years | Oct 1978 - Aug 1987 | Jun 1987 - 2006 | Sep 2014- |

**Table 2.** The major characteristics of the passive microwave imagers, directly relevant to this study. The instruments include other channels but are not used here. *failure of the 21 GHz channel in 1985 and drift from 1986.

and Draper (2016). In addition to the usual hot load, GMI uses noise diodes improving calibration accuracy. The satellite can also perform flight maneuvers to correct drift and improve calibration. Lean (2017) found low biases for all channels, as compared to ECMWF simulations (lower than 0.8 K). All these technical specificities make GMI an excellent reference for inter-calibration purposes. In this study, the calibrated $T_b$ Level 1C (Hou et al. 2014) are used.

5  The first SSM/I was launched in 1987 on-board the F08 satellite of the Defense Meteorological Satellites Program (DMSP). The SSM/I instruments were upgraded to SSMIS in 2003, with more channels for atmospheric profiling of temperature and water vapor. Both will be referred to as the SSM/I series here. The DMSP polar orbiters cannot correct orbital degradation and as a consequence, the instruments are subject to drifts in the overpassing times, making instrument inter-calibration more challenging. Here, the SSM/I data provided by the CM SAF without the intercalibration layer will be used to evaluate the

10 consistency of our SMMR land calibration.

## 2.2 The method

SMMR and GMI do not have any common observing period. Therefore we need to rely on a statistical analysis of SMMR and GMI observations from different years to provide a correction of the SMMR biases over continental surfaces. The fundamental hypothesis here is that the changes in the environmental conditions affecting the microwave signals are limited between the

15 1980s and 2010s, as compared to the SMMR and the GMI instrument calibration differences.

  The SMMR and GMI EIA difference is ~3°. Over ocean, this can strongly affect the signal, due to the sensitivity of the ocean emissivity to the incidence angle as well as to the changing atmospheric contribution with angle. Over land, the surface emissivity is usually high for both polarizations, with values of 0.9 and higher for most surface types (Prigent et al. 2006), with very limited changes with EIA (Prigent et al. 2000). With a high land surface emissivity, the atmospheric contribution to the

20 signal is small compared to the surface one and the changes in the atmospheric contribution with angle will not significantly affect the measurement. By the same token, the land surface emissivities being rather high and showing very smooth variations





with frequency (Prigent et al. 2000), the differences in frequencies between the Ku and Ka channels of SMMR and GMI is not expected to significantly affect the signals.

To facilitate the comparison, each dataset is projected on the Equal Area EASE 2.0 grid globally between 60°S and 70°N and on the EASE southern hemisphere azimuthal grid over the Antarctica area (Brodzik et al. 2012). A land mask is used to
filter out water pixels at the 25 km resolution.

GMI data are collected for two months in July and August (respectively January and February) from 2015 to 2017. This allows to reproduce the boreal summer and winter $T_b$ diurnal cycles. As GMI has a higher spatial resolution than the target grid, all observations falling within a grid cell of ~12 km radius are averaged to derive one value per grid cell. Regarding the temporal dimension, the values are averaged with a 15 minutes window for each grid cell to suppress small scale variability in
the signal. If no observation in a given grid cell is available in a particular 15 minutes interval, a linear interpolation with the closest existing measurements is performed. Finally, to get a smooth diurnal cycle, a moving average is applied over each grid cell with a 75 minutes window (two data points before and after the target one).

For SMMR, $T_{bs}$ are collected from 1981, 1982, and 1987 for the same months as GMI. The observations are available twice a day (for the ascending and descending overpasses). Given the comparable resolutions of SMMR observations and the
target grid, a nearest-neighbour technique is used to project the $T_{bs}$ into the EASE grid. Using several years of data for each instrument can alleviate possible effects of strong inter-annual variability in the signals (related to Niño or Niña events for instance).

The resulting datasets contain almost 2 million data points spanning all the continents and two contrasted seasons (summer and winter).

## 3  Results

### 3.1  SMMR and GMI comparisons at regional scale

At regional scales, a set of different areas have been selected to represent a large variability in environmental conditions. The averaged $T_{bs}$ diurnal cycle derived from GMI is calculated for sample areas covering 25 grid cells, over two months, over 3 years (2015-2017). For the same areas, the averaged SMMR $T_{bs}$ for the ascending and descending orbits are also calculated,
over the same months, for 1981, 1982 and 1987.

Figure 1 presents the results for areas in the North Hemisphere, during winter (January and February), at 18 and 37 GHz, for both orthogonal polarizations. It includes cold mountainous regions, arid deserts with high $T_{bs}$ during the day and a large amplitude of the diurnal cycle, as well as warm rainforest with a limited variation in $T_{bs}$. Forested regions show a low diurnal cycle amplitude of the $T_{bs}$ (<10 K), with almost no difference between V and H polarizations due to the emission and scattering
effect of the dense vegetation. Arid regions (cold or warm) have large diurnal variations in $T_{bs}$, directly related to the diurnal cycle of the land surface temperature. The polarization difference is significant, due to an almost specular behavior of these rather flat surfaces. With increasing vegetation cover (sparsely vegetated or grassland) an intermediate behavior is observed. The variability associated to each GMI average is computed and also displayed (the grey shades around the diurnal cycle



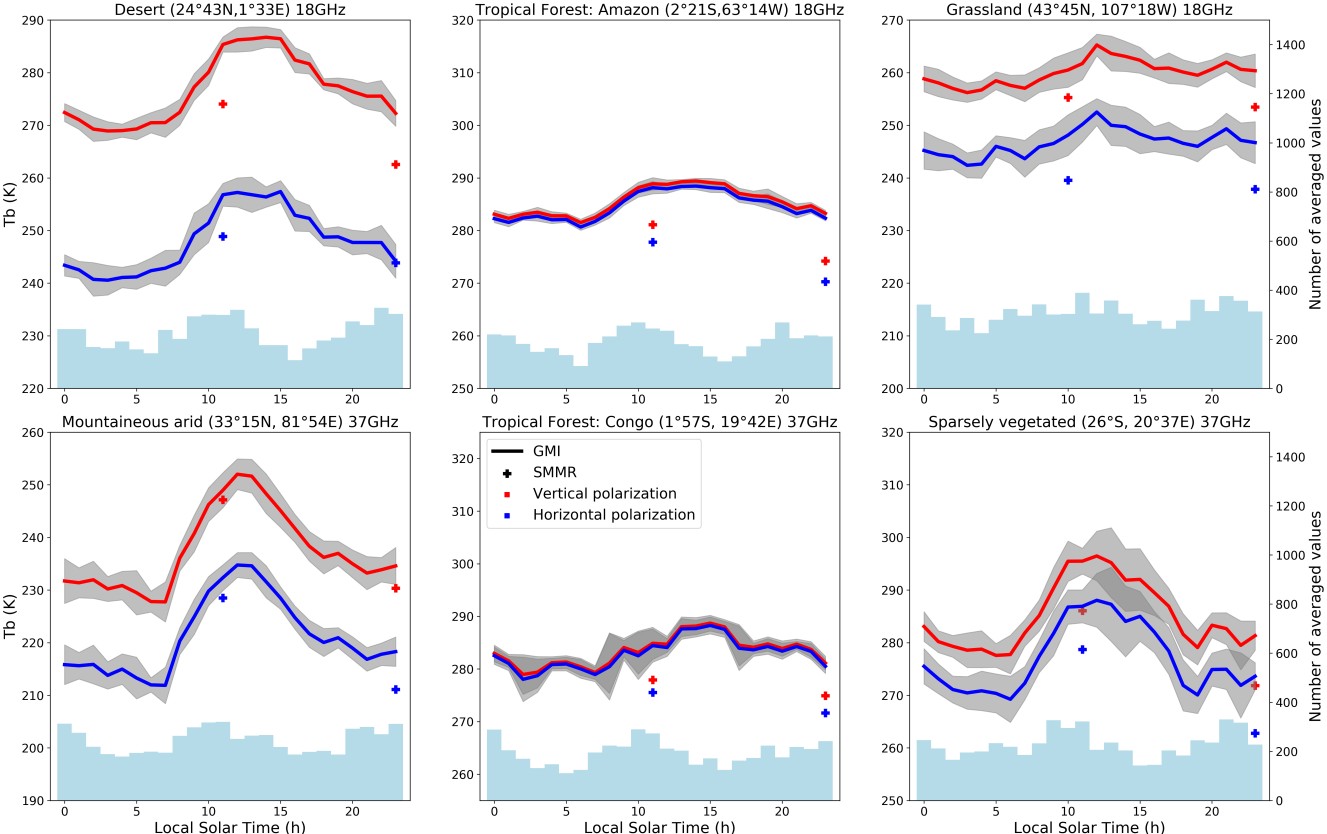

**Figure 1.** Diurnal cycles from GMI compared to SMMR values at different frequencies and locations averaged for the January and February months (horizontal polarization in blue and vertical in red). Measurements are averaged over 25 grid cells and by hour with the standard deviation of each average temperature in the diurnal cycle and the associated number of measurements displayed below.

indicate one standard deviation). The number of individual pixels used in the diurnal cycle calculation is also indicated (blue shades). The low variability of the signals over the Amazon confirms the high stability of the $T_{bs}$ in this region that is regularly used as a warm radiometric reference (e.g., Brown and Ruf (2005)). The SMMR values have been calculated for the initial Level 1B data as well as for the CM SAF inter-calibrated results but only the level 1B data is displayed on Fig. 1. Note that the
5    CM SAF inter-calibration is not designed nor validated for observations over land.

The SMMR day and night values follow the same diurnal pattern as GMI, but with a systematic underestimation of the $T_{bs}$, regardless of the frequency and polarization. These differences are usually significantly larger than what is expected from the GMI signal variability, indicated by the standard deviation around the average. The difference between SMMR and GMI appears to be almost the same for both polarizations. The difference seems to decrease for the coldest locations. The
10    variability is higher under mid latitudes where year-on-year variations are possible and meteorological variations can impact the measurements. The differences cannot be explained solely by the instrument different characteristics (EIA or frequencies),



or by environmental changes between the periods covered by the two instruments (the 1980s for SMMR and the 2010s for GMI). A possible source of error could be an erroneous warm calibration load temperature.

The inter-calibration layer from CM SAF for SMMR does not seem to improve the results. Over land the mean correction added by the calibration is below 0.5 K for the 18 GHz vertical and horizontal polarization and 37 GHz horizontal channel.

5 The correction for the 37 GHz vertically polarized channel is on average around 2 K. The inter-calibration has been designed mainly for ocean observations with low $T_{bs}$ and not for our land application so it does not adequately correct for calibration issues over land (not shown on the figure). Other locations and seasons were analyzed, with similar conclusions (not shown here).

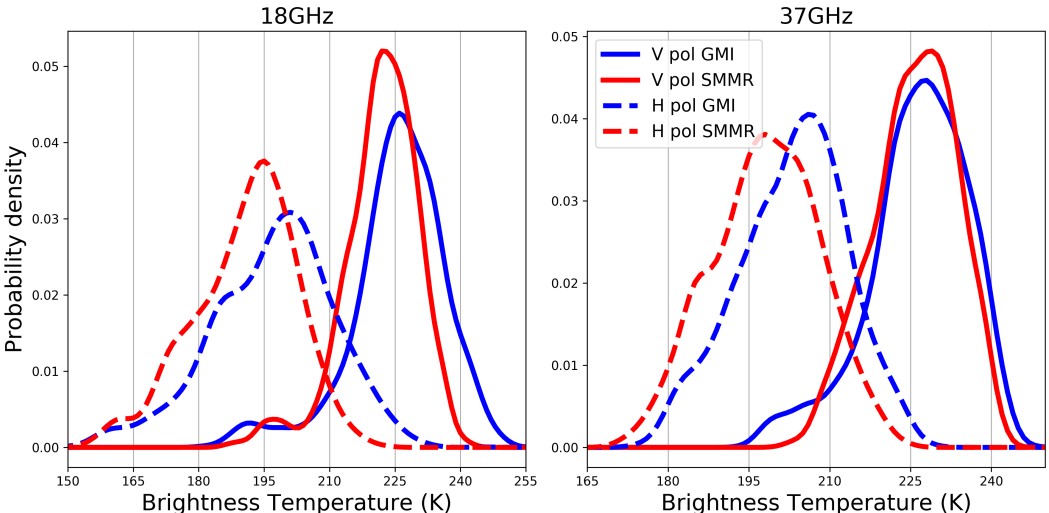

**Figure 2.** Probability density functions of July/August brightness temperatures over Antarctica for GMI in 2015 and SMMR in 1987.

The observations over the Antarctic ice sheet are also explored, to extend our investigation to lower $T_{bs}$. Because of the

10 GMI coverage, the comparison is limited to the edge of Antarctica. Care is exercised to avoid contamination by the ocean and sea ice. South Hemisphere winter months are selected (July and August). During this long night, there is no diurnal cycle and the $T_{bs}$ distribution for the two satellites are directly compared (Figure 2). The delta between SMMR level 1B and GMI $T_{bs}$ distribution is between 5 and 10 K for all channels except the 37 GHz vertical polarization that shows a lower difference.

Here as well, the significant differences in the SMMR level 1B and GMI $T_{bs}$ cannot be attributed only to changes in

15 environmental conditions, even over ∼30 years. Jezek et al. (1991) compared SMMR and SSM/I over the Antarctic ice sheet during their overlap period and also found significant differences in both Ku and Ka bands. They discussed the impact of the change in EIA and frequency between the instruments but concluded that it cannot explain the large observed differences. Only calibration issues can explain the observed differences, with the SMMR $T_{bs}$ colder than the GMI ones.





## 3.2 Derivation of a SMMR correction over continental surfaces

In order to assess the possibility to correct for the SMMR calibration issue, SMMR and GMI observations are compared at global scale. The SMMR observations (both ascending and descending orbits) are compared to the corresponding GMI values for the same times in the day, for January and February and for July and August, over 3 years for the two instruments (1981-1982-1987 for SMMR and 2015 to 2017 for GMI, as before). Points for which the difference between the GMI and SMMR values is outside 3 standard deviations from the mean difference are suppressed. We checked that these points were essentially located over coastal regions: GMI has a better spatial resolution than SMMR and over the coasts the SMMR observations will likely include more contributions from the surrounding ocean. Figure 3 displays the distribution of the GMI against the SMMR $T_{bs}$, at 18 and 37 GHz, both polarizations. The underestimation of the SMMR $T_{bs}$ compared to the GMI ones clearly increases with $T_{bs}$. Here we suggest a simple linear correction to inter-calibrate the SMMR observations toward the GMI ones. A more sophisticated correction would not be justified. First, we are aware that this is a first order inter-calibration, as the comparison involves different years. Second, it would be more complex to implement. Lastly, it would likely overfit some part of the signal rather than correcting the $T_{bs}$.

A simple correction is proposed for all channels with the form $\hat{T}_{SMMR} = a \times T_{SMMR} + b$. The coefficients slope $a$ and the intercept $b$ are estimated through the minimization of the sum of squared difference : $\sum_{i=1}^{N}(T_{GMI} - a \times T_{SMMR} - b)^2$ derived from the assumption that $\hat{T}_{SMMR} = T_{GMI}$, with $N$ the number of data points used for the regression coefficient estimation. The data collection covers the full $T_b$ range observable over continental surfaces, including some polar regions.

The distribution of the data points from the cold and warm ends is uneven, with less points for lower $T_{bs}$. To alleviate this issue, we randomly sample the points from both the cold and warm ends to simulate an even distribution over the full $T_b$ range. Different samplings were tested to confirm the stability of the estimated coefficients. The resulting regression lines are added to Fig. 3. The mean squared errors of the linear regression are indicated with grey shades. It appears clearly that the uncertainty on the areas with less data (between the Antarctica temperatures and the land data points) or near the edge of the range of values are the ones with the highest errors. The coefficients of the linear regressions are provided in Table 3. The uncertainties on the regression coefficients are computed with a t-test for a 99% confidence level.

| Channel | Slope | Intercept | $R^2$ |
|---------|-------|-----------|-------|
| 18V | 1.10±0.01 | -18.7±2.2 | 0.976 |
| 18H | 1.05±0.01 | -1.29±1.9 | 0.971 |
| 37V | 1.15±0.01 | -32.2±2.2 | 0.976 |
| 37H | 1.04±0.01 | -1.23±1.9 | 0.976 |

**Table 3.** Estimated linear correction coefficients, for the inter-calibration of the SMMR observations at 18 and 37 GHz, vertical and horizontal polarizations, with respect to GMI observations over the continents. The uncertainties are also added (computed with a t-test at 99% confidence level).



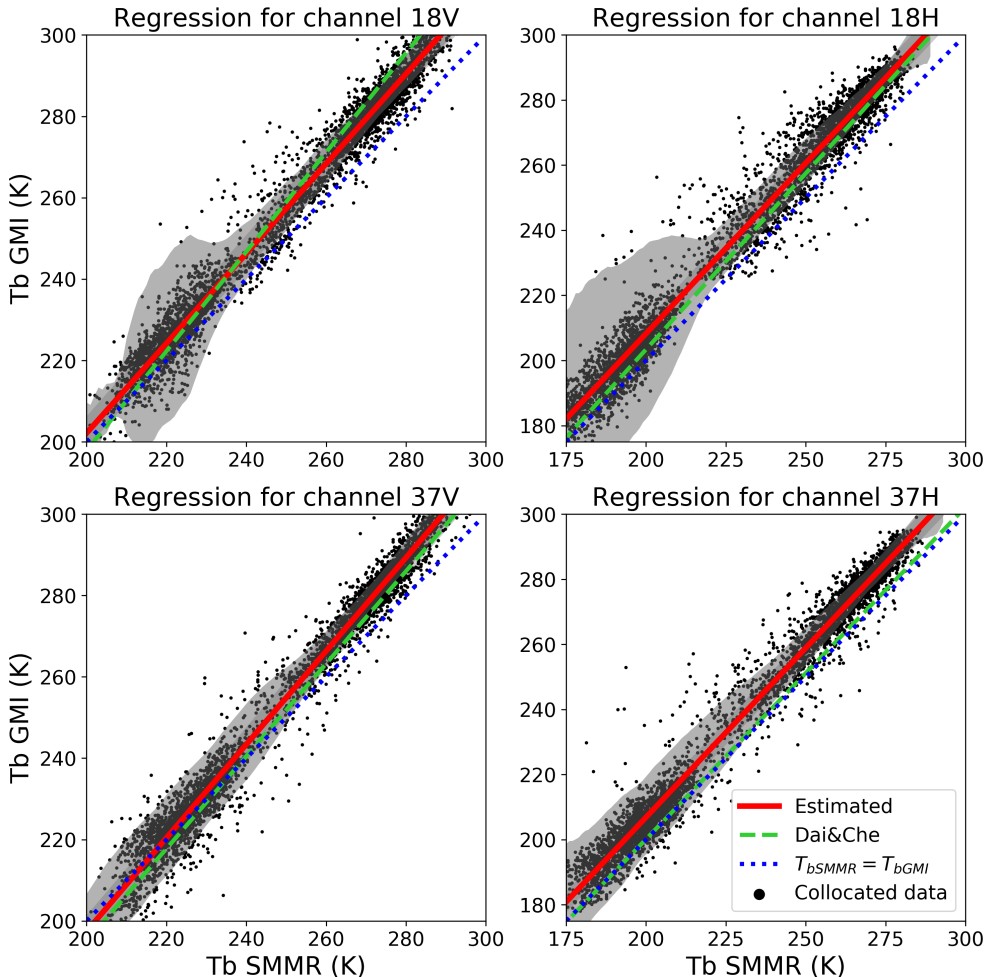

**Figure 3.** Linear regression on SMMR values against GMI after filtering points. In addition the correction derived by Dai and Che (and corrected for the SSM/I F08 to GMI calibration) is displayed.

## 3.3 Evaluation

The suggested corrections are applied to SMMR values and the results can be compared to other imager observations. The inter-calibration being derived from GMI, a good agreement is expected with that sensor. The first consistency check consists in comparing the calibrated SMMR observations to the SSM/I observations, for their overlapping period. The GMI-derived diurnal cycle of $T_{bs}$ is used as a bridge between the SMMR and SSM/I observations that have different overpassing times. Figure 4 presents the comparisons of SMMR and SSM/I F08, for different locations, averaged over their common period in July and August 1987, along with the GMI-derived diurnal cycle of the $T_{bs}$ (estimated over three different years). It shows that SSM/I $T_{bs}$, used without any inter-calibration, are in good agreement with the $T_b$ diurnal cycle estimated from GMI.





Contrarily to the SMMR observations before inter-calibration, no obvious large bias is observed between SSM/I and GMI, even for the very warm scenes, at 18 and 37 GHz for both polarizations. The average difference between SSM/I F08 (without any inter-calibration scheme applied) is around 2.5 K for the 18 GHz channels, and around 0.5 K for the 37 GHz channels. A bias is expected between measurements that have not been inter-calibrated, but it is lower than the one detected for SMMR and

5 confirms our assumptions regarding the emissivity behavior over continental surfaces as well as the small variation in the 30 year gap between the 1980s and the 2010s.

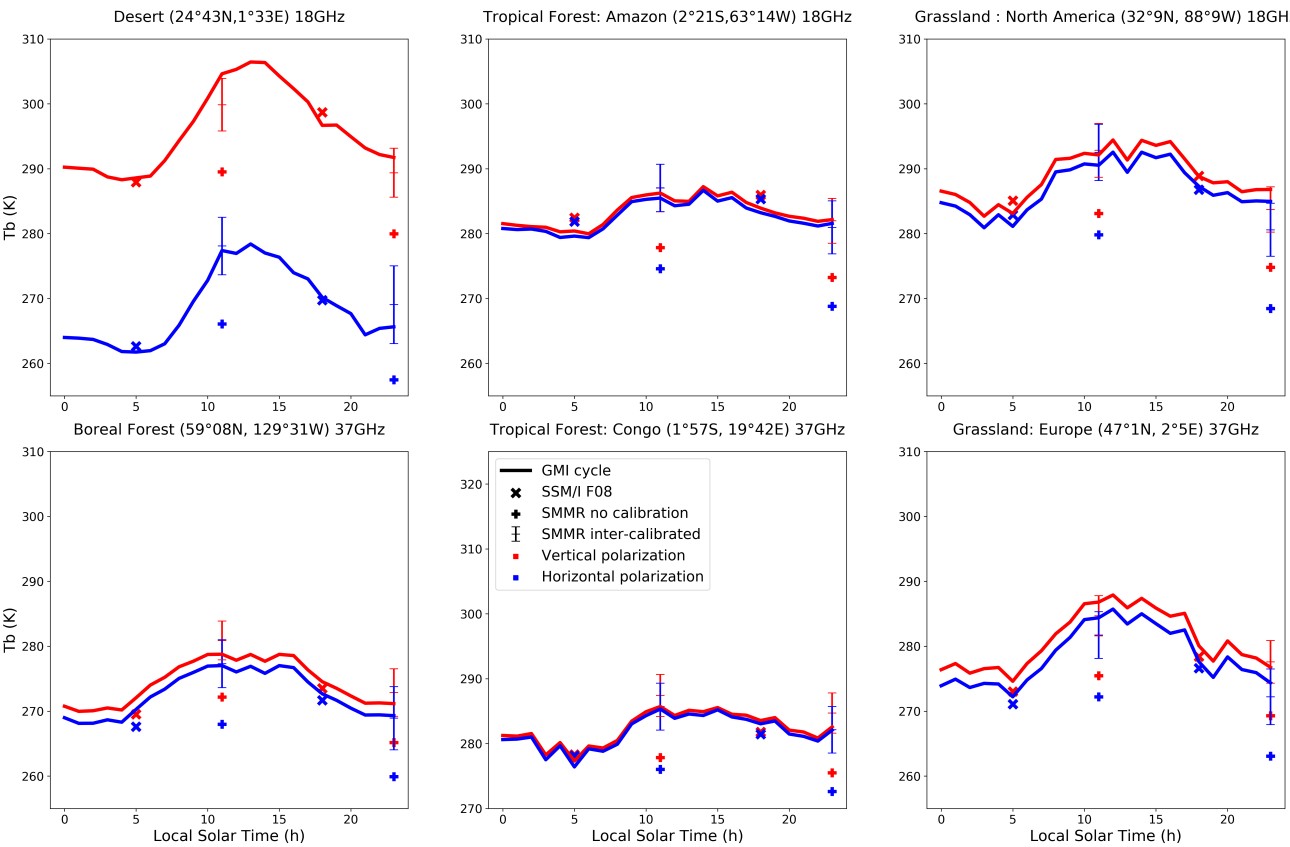

**Figure 4.** Diurnal cycle of GMI in July/August compared to values measured by SMMR with and without the proposed correction, and SSM/I values for the months of July/August 1987 over various locations and at different frequencies.

The agreement between the SMMR inter-calibrated values and GMI is clearly improved, as compared to the previous SMMR Level 1B results. These results show that the SSM/I F08 and the newly inter-calibrated SMMR are consistent, at least within the uncertainty provided with this inter-calibration. More cases have been tested, with similar results.

10 In addition to the evaluation with other sensors, the results of our inter-calibration are compared with an alternative method developed for SMMR. Dai and Che (2009) built their inter-calibration scheme upon observations in desert and polar areas. The SSM/I F08 instrument is adopted as the reference and a model is developed to account for the diurnal changes in temperatures.





The resulting linear regressions are added on Fig. 3 (green dotted lines). The two inter-calibration methods are based on very different principles. Nevertheless, they agree well for all channels, with similar slopes and slightly different intercepts. This adds confidence to our methodology.

## 4   Conclusions

Several FCDRs from passive microwave imagers have been produced, from pioneer instruments such as SMMR to the AMSR series. So far, the efforts essentially focussed over ocean where radiative transfer simulations fed by reanalysis can serve as a reference to bridge the differences between the instruments, in terms of frequency, EAI, and overpassing time. Over continental surfaces, inter-calibration of passive microwave imagers is challenging, especially for Sun-synchronous polar orbiting satellites with different over-passing times at the equator. Here we derived a method to extend the FCDR collection to SMMR over land

at 18 and 37 GHz, using the non Sun-synchronous GMI instrument as a reference, despite the lack of a common observing period. GMI observations are used to reconstruct the diurnal cycles of $T_{bs}$ that should be observed with SMMR, with the assumption that the environmental conditions have not changed drastically over the last 30 years and that the differences in frequencies and EIA between the two instruments can be neglected over land. With these hypotheses, the objective is to correct for the large differences between the sensors. Before inter-calibration, the comparison of the SMMR and GMI observations

show a significant underestimation of the $T_{bs}$ with SMMR, and this underestimation tends to increase with increasing $T_{bs}$. A linear regression is suggested for the 18 and 37 GHz channels, vertical and horizontal polarizations, to calibrate the SMMR observations toward the GMI estimates. The SMMR correction is evaluated with respect to the SSM/I F08 observations over their overlapping period in orbit, in July and August 1987. The GMI-derived diurnal cycle of $T_{bs}$ acts as a bridge between the two instruments that have different overpassing times. A good agreement is reached between all sensor measurements. This

inter-calibration of the SMMR instrument over land will make it possible to extend the passive microwave estimations of land surface variables over 9 more years backward, from 1978 to 1987. This will be practically tested in the near future for the estimates of e.g., surface water extent and land surface temperatures from microwave observations, two variables that rely on the 18 and 37 GHz observations for their retrieval, under clear and cloudy sky conditions.

*Data availability.*  The Satellite Application Facility on Climate Monitoring provides access to the Fundamental Climate Data Record of

Microwave Imager Radiances (10.5676/EUM_SAF_CM/FCDR_MWI/V003; Fennig et al. (2019)). The Global Precipitation Measurement Mission Microwave Imager GMI_R Common Calibrated Brightness Temperatures Collocated L1C 1.5 h 13 km V05 (GPM_1CGPMGMI_R 10.5067/GPM/GMI/R/1C/05; Berg et al. (2016)) is provided by NASA.

*Author contributions.*  All authors have been involved in interpreting the results, discussing the findings, and editing the paper. SF conducted the main analysis and wrote the draft of the paper. CJ and CP provided guidance on using the data sets and expertise on analysing the results.



*Competing interests.* The authors declare that they have no conflict of interest.

*Acknowledgements.* The authors would like to acknowledge the data providers: Wesley Berg and Christian Kummerow from Colorado State University for the discussion on their intercalibration method, and special thanks are due to Karsten Fennig and Marc Schröder for their help on using the CM SAF FCDR and the comments on the work done.





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
