# Peer review of "Inter-calibrating SMMR brightness temperatures over continental"

_Atmospheric Measurement Techniques, 2019_

## Referee Comment (RC1) · Anonymous Referee #1 · 3 Apr 2020

The authors have developed a method for adjusting the calibration of the Scanning Multichannel Microwave Radiometer(SMMR) on the Nimbus 7 satellite. This instrument was plagued with a multitude of problems making it very difficult to work with. However, problematic or not, it was the only microwave imager operating for almost a decade in the 1970s and 1980s. It is incumbent on the scientific community to salvage the data from this instrument as well as possible. The present paper is a significant contribution to this effort.

In spite of the lack of a common observing period, they have chosen to use the GPM Microwave Imager (GMI) on the Global Precipitation Measurement satellite as a reference. The GMI is exceedingly well calibrated and is suitable as a reference instrument. They have to make assumptions in order to get around the lack of a common observation period.

They have only dealt with two of the five frequencies on SMMR. The radio frequency interference problem would make any use of the 6.6 GHz channel very difficult and the 21 GHz channel is not very useful over land. However, it is disappointing that they did not include the 10.7 GHz channel in their study.

The writing is not exactly native English but there is no problem with understanding.

Detailed comments:

P4 Line20: "Njoku et al. (1980)" This reference is for the SMMR on SeaSat which only lasted for 99 days. It is of limited applicability to the SMMR on Nimbus 7 which is the topic of this paper.

P5 Line 6: "...upgraded to SSMIS..." It's not good to lump the SSM/I and the SSM/IS together. They were manufactured by different companies and were very different in terms of the problems. In particular, the SSM/IS had a very large problem with emission from the main reflector. From a calibration point-of-view they are not at all the same sensor.

P5 Line 13 "The fundamental...changes in the environmental conditions..." This is a necessary assumption for them to proceed, but it is also a severe limitation. The results cannot be used to look at secular changes over the 3 decade time difference between the two satellites. While this is a seemingly obvious limitation, they should highlight it. Otherwise somebody will waste a lot of time and effort drawing specious conclusions.

P6: They compare the various channels of SMMR and GMI directly with no algorithm to account for small frequency and view angle differences. They argue that these differences are small. Given the problems of the SMMR, these differences are probably small relative to the other uncertainties in the comparison. However, for comparisons of higher quality sensors (e.g. Windsat vs GMI), this would not be adequate. When I agreed to review this paper, I was hoping that I would see some land surface modeling

to support the intercomparison. Alas, 'twas not to be.

P8 Line 2. "...erroneous warm calibration load temperature" Note that an error in correcting for the portion of the antenna pattern that misses the Earth would have the same form. Either one would result in an intercept of 2.7K and only slightly different slopes than given in Table 3.

———————————————————

---

## Referee Comment (RC2) · Linwood Jones (Referee) · 6 Apr 2020

Linwood Jones (Referee)

wlinwoodjones@gmail.com

This paper presents the results of an inter-satellite radiometric calibration for SSMR over land using the GPM instrument. The authors' approach is novel in that they perform the inter-calibration of two satellite radiometers without near-simultaneous collocated observations. This method is justified because the SMMR was the first conical scanning radiometer in space, and therefore did not have the usual over-lapping period for comparisons with other space-borne instruments. Given this situation, I feel that the authors have made a reasonable case for their statistical method developed to compare the current GMI with the previous SSMR.

However, I have some comments, which I feel would strengthen their case if included

in their paper.

1) Concerns the three available SMMR Tb datasets that exist. In section 2.1 the CM SAF FCDR was described but not the others? Were they available? It would have been better to compare the three different data sets in their statistical analysis or at the very least to discuss why they were not considered.

2) Concerns the oceans and/or Antarctic sea ice data sets. I suspect that a similar statistical comparison could have been made (as presented for land). I recognize that this expands the scope of the analysis, but it also makes the paper stronger. I suspect that similar results would have been found, which would provide confidence to the conclusions.

3) Concerns the selection of the two 2-month periods, namely: Jan/Feb and Jul/Aug. Some discussion was provided in section 2.2, but I recommend more information be provided to inform the reader specifically why these were selected (as opposed to monthly comparisons)?

4) The SMMR biases, relative to GMI (SSMI), are presented in Fig-1, -2 & -4, but I recommend that they also are included in a Table of results.

The following corrections are suggested:

P-2 Line-3 following Seasat insert "Nimbus-G"

P-2 same pp WindSat and TMI were not mentioned in the list of radiometers? Since only SSMI and GMI were involved in the direct comparisons, the others could have been omitted?

P-2 Line-12 . . . Fennig et al. 2019) than that . . .

P-2 Line-25 Insert "However", this method is very

P-4 Line-5 . . . CM SAF FCDR insert "SMMR" . . .

---

## Author Comment (AC1) · 17 Apr 2020

The authors would like to thank the anonymous reviewer for the comments on the paper "Inter-calibrating SMMR brightness temperatures over continental surfaces". Indeed the Scanning Multichannel Microwave Radiometer (SMMR) is an important piece in the story of earth radiometric measurement that can still be useful for the remote sensing community.

Response to the review

The authors have developed a method for adjusting the calibration of the Scanning Multichannel Microwave Radiometer(SMMR) on the Nimbus 7 satellite. This instrument

was plagued with a multitude of problems making it very difficult to work with. However, problematic or not, it was the only microwave imager operating for almost a decade in the 1970s and 1980s. It is incumbent on the scientific community to salvage the data from this instrument as well as possible. The present paper is a significant contribution to this effort. In spite of the lack of a common observing period, they have chosen to use the GPM Microwave Imager (GMI) on the Global Precipitation Measurement satellite as a reference. The GMI is exceedingly well calibrated and is suitable as a reference instrument. They have to make assumptions in order to get around the lack of a common observation period. They have only dealt with two of the five frequencies on SMMR. The radio frequency interference problem would make any use of the 6.6 GHz channel very difficult and the 21 GHz channel is not very useful over land. However, it is disappointing that they did not include the 10.7 GHz channel in their study. The writing is not exactly native English but there is no problem with understanding.

**Response** The 10.7 GHz channel was not included in the study for two main reasons: First the goal to provide continuous times series of measurements cannot be achieved with this channel as the SSM/I and SSMIS radiometers do not include it. Therefore a 40 year series of 10.7GHz measurements is not possible. Second, the 10.7GHz channel is not used in our retrieval of the Land Surface Temperature that was the initial goal of this study. It could be studied in a similar fashion as GMI also has a channel at 10.65GHz (against 10.7GHz for SMMR).

The authors have tried to improve the writing to better fit the English standard.

Detailed comments

P4 Line20: "Njoku et al. (1980)" This reference is for the SMMR on SeaSat which only lasted for 99 days. It is of limited applicability to the SMMR on Nimbus 7 which is the topic of this paper.

**R** The reference was mixed up with another paper by E. Njoku in 1980. It will be replaced with the following "Antenna pattern correction procedures for the Scanning Multichannel Microwave Radiometer (SMMR)" E. Njoku 1980.

P5 Line 6: "...upgraded to SSMIS..." It's not good to lump the SSM/I and the SSM/IS together. They were manufactured by different companies and were very different in terms of the problems. In particular, the SSM/IS had a very large problem with emission from the main reflector. From a calibration point-of-view they are not at all the same sensor.

**R** Although the two instruments series differ on many points they still share some common characteristics such as the channels used, this was the intended meaning of this sentence. The updated sentence will make the distinction between these instruments more apparent. p.5 l.6 "In the following years multiple instruments were launched such as the Tropical Rainfall Measuring Mission's (TRMM) Microwave Imager (TMI) in 1997, or the Special Sensor Microwave - Imager/Sounder (SSMIS)"

P5 Line 13 "The fundamental...changes in the environmental conditions..." This is a necessary assumption for them to proceed, but it is also a severe limitation. The results cannot be used to look at secular changes over the 3 decade time difference between the two satellites. While this is a seemingly obvious limitation, they should highlight it. Otherwise somebody will waste a lot of time and effort drawing specious conclusions.

**R** As the reviewer noticed it is a very important assumption for us to proceed with the method. And as it is it should prevent any one from trying to perform trend analysis using the corrected SMMR data. The limitation has been more clearly stated to leave no doubt. p.12, l.22: "However, given the use of the more recent GMI instrument as a calibration reference any comparison between different epochs should be conducted with extreme care."

P6: They compare the various channels of SMMR and GMI directly with no algorithm to account for small frequency and view angle differences. They argue that these

differences are small. Given the problems of the SMMR, these differences are probably small relative to the other uncertainties in the comparison. However, for comparisons of higher quality sensors (e.g. Windsat vs GMI), this would not be adequate. When I agreed to review this paper, I was hoping that I would see some land surface modeling to support the intercomparison. Alas, 'twas not to be.

**R** The change of frequency and incidence angle are the two major differences between the two instruments. The change of surface emissivity with regard to these parameter has not been modeled but it could be done if trying to inter-calibrate other MW instruments that have a more reliable calibration.

P8 Line 2. "...erroneous warm calibration load temperature" Note that an error in correcting for the portion of the antenna pattern that misses the Earth would have the same form. Either one would result in an intercept of 2.7K and only slightly different slopes than given in Table 3.

**R** That is a good point, the sentence was changed to take into account this possible error source as well. p8, l1: "Different sources could cause such errors, for instance an erroneous warm calibration load temperature or an error in the correction of antenna pattern that misses the Earth."

Again the authors are grateful for the very good review that offered improvement to various parts of the paper as well as correcting some mistakes. The overall quality of the paper has been improved after taking into account these comments.
* * *

---

## Author Comment (AC2) · 17 Apr 2020

The authors are grateful for the review by Pr. Linwood Jones of the paper "Inter-calibrating SMMR brightness temperatures over continental surfaces". The comments were specific and definitely helped the authors improve the paper for future readers.

**1   Response to the review**

This paper presents the results of an inter-satellite radiometric calibration for SSMR over land using the GPM instrument. The authors' approach is novel in that they perform the inter-calibration of two satellite radiometers without near-simultaneous collo-

cated observations. This method is justified because the SMMR was the first conical scanning radiometer in space, and therefore did not have the usual over-lapping period for comparisons with other space-borne instruments. Given this situation, I feel that the authors have made a reasonable case for their statistical method developed to compare the current GMI with the previous SSMR. However, I have some comments, which I feel would strengthen their case if included in their paper.

1) Concerns the three available SMMR Tb datasets that exist. In section 2.1 the CM SAF FCDR was described but not the others? Were they available? It would have been better to compare the three different data sets in their statistical analysis or at the very least to discuss why they were not considered.

**Response** : We only described the dataset released by the CM SAF as it is the one we used. The reason for that choice is twofold : First the CM SAF released the original data as well as their inter-calibration layer, making it easy to use an uncalibrated dataset. The unprocessed dataset would have been the same from any of the other providers. We still compared the inter-calibration layer from CM SAF as it was available with adequate documentation. Furthermore, the contact with CM SAF researcher was easy and they helped us through various steps in using their dataset.

2) Concerns the oceans and/or Antarctic sea ice data sets. I suspect that a similar statistical comparison could have been made (as presented for land). I recognize that this expands the scope of the analysis, but it also makes the paper stronger. I suspect that similar results would have been found, which would provide confidence to the conclusions.

**R** The study of sea-ice datasets is indeed a very important topic, and it could be the subject of a specific study. However, the need for inter-calibration for the retrieval of sea ice parameters is more limited, as most sea ice concentration algorithms include inter-calibration processes, with the use of what is called dynamic tie-points. Within the algorithms, changes in the instrument responses is already taken into account

(e.g., Tonboe et al., 2016). The ocean comparison was rapidly tested and the results showed that the methodologies from the different centers yield similar results.

3) Concerns the selection of the two 2-month periods, namely: Jan/Feb and Jul/Aug. Some discussion was provided in section 2.2, but I recommend more information be provided to inform the reader specifically why these were selected (as opposed to monthly comparisons)?

**R** The two periods selected (6 months apart) offer a large sample of possible Earth states, that are needed for a robust statistical comparison. A month by month comparison could have also been done, but it would most likely only add complexity to the paper without improving the proposed inter-calibration. Section 2.2 was updated to improve readers understanding of the underlying choice in the methodology.

4) The SMMR biases, relative to GMI (SSMI), are presented in Fig-1, -2  -4, but I recommend that they also are included in a Table of results. **R** The Figures 1,2 and 4 only show SMMR biases on specific locations. A global evaluation of the bias is indirectly included in Table 3, with the temperature dependant correction.

**2  Specific corrections**

The following corrections are suggested:

P-2 Line-3 following Seasat insert "Nimbus-G" **R** Added to the corrected paper

P-2 same pp WindSat and TMI were not mentioned in the list of radiometers? Since only SSMI and GMI were involved in the direct comparisons, the others could have been omitted? **R** Indeed the other radiometers are not used and have been removed from the paper.

P-2 Line-12 . . . Fennig et al. 2019) than that . . . **R** Corrected in the paper

P-2 Line-25 Insert "However", this method is very **R** Sentence changed.

P-4 Line-5 . . . CM SAF FCDR insert "SMMR" . . . **R** Corrected in the paper.

The authors appreciate the thorough review by Pr. Linwood Jones that offered precise comments to improve possible unclear areas in the paper. We hope that this answer will remove any doubts concerning the methodology and choices made in the study.

---

## Referee Report (RR1)

Manuscript:    AMT-2019-493
Title:         Inter-calibrating SMMR Brightness Temperature over Continental Surfaces
Authors:       Samual Favrichon, Carlos Jimenez and Catherine Prigent

**Recommended Disposition:** Accept with minor revisions

**General Comments:**

I initially had some serious reservations regarding the authors attempt to intercalibrate SMMR
Tbs using data from GMI almost 30 years later. After reading the manuscript, however, I think
that the authors are providing some valuable insight into the SMMR data, although I would like
to see them strengthen their conclusions a bit regarding the limitations of the SMMR data. I tend
to agree that the SMMR data has value for certain applications, but any attempt to use this data
to look at climate trends or more subtle variations is definitely not warranted. I believe the
authors conclusions bear this out, but I would like to see a bit more explicit statement to this
effect.

**Specific Comments:**

Page 3, line 17: The statement "Here, we suggest to use GMI as a reference instrument,
assuming that the environmental conditions have not changed drastically from the SMMR to the
GMI era, to allow the comparison of a large set of observations averaged over time. This strategy
does not allow to perform a detailed intercalibration but it makes it possible to correct for major
biases that so far hamper the use of SMMR over land for the generation of climate record of
geophysical parameters." This statement regarding the nature of the intercalibration needs to be
reiterated in the conclusions. I'm also a bit concerned with the statement suggesting that the
intercalibrated data can subsequently be used for climate data records. While it depends greatly
on the specific application, certainly using SMMR data to extend the global satellite time series
for identifying subtle climate trends is not warranted.

Page 5, Table 2: GMI has operated from March 2019 forward. The data have been reprocessed
back to this date to provide a consistent data record.

Page 5, line 13. Again, this hypothesis of limited changes in environmental conditions comes
with significant caveats given climate change.

Page 6, line 4: This statement is a bit unclear. Do the authors mean GMI data are collected for 2
months in summer (i.e. July and August) and two months in winter (i.e. January and February)?

Page 6, line 12: Did the authors attempt to look at potential calibration differences between the
three years of SMMR data used (i.e. 1981, 1982 and 1987)? Is there any indication of a time-
dependent change or calibration drift (or lack thereof)?

Page 7, figure 1: I'm a bit unclear on the location and extent of the regions chosen. The text
mentions 25 grid cells. How large is the region covered by 25 grid cells. A figure showing the
various regions selected for the results in Figure 1 would be useful.

Page 8, line 1: The relative consistency in the slopes shown in Figure 3 does lead point to the possibility of an error in either the warm load temperature or the antenna pattern correction (i.e. spillover). This might be worth mentioning in the conclusion, although the results by Dai and Che don't appear as consistent between channels.

Page 8, Figure 2: The large difference between the histograms shown over Antartica (i.e. 5-10K for all except the 37 GHz V-Pol channel) raises some questions. Is this result consistent with the subsequent results shown in Figure 3? The authors mention screening out ocean and sea ice, but isn't the land region covered by snow/ice? What impact might changes in snow/ice emissivity have on the observed Tb? Emissivity of snow/ice is difficult to determine and could potentially be influenced by a number of different factors and thus have little or nothing to do with calibration differences. As a result, unless these differences are consistent with the linear fit derived from Figure 3, it isn't clear to me that these results over Antartica are useful. Please justify.

Page 9, Table 3: The values of slope and intercept from the linear fit don't provide the reader a clear sense of the magnitude of the calibration differences. I suggest that the authors consider adding the mean intercalibration difference and uncertainty for GMI Tb values of 200 and 300K. This would provide the reader a much better sense of the magnitude of the adjustment as well as the uncertainly for both cold (ocean) and warm (land) scenes.

Page 12, line 13: I would like to see the authors strengthen and better qualify the statement "With these hypothesis, the objective is to correct for the large differences between the sensors". As mentioned regarding Table 3, exactly how large are these differences for typical cold ocean and warm land scenes (i.e. 200 and 300 K)? Ultimately, I believe that highlighting the limitations of the SMMR data is as important if not more so than that value of the corrected data itself. The results in Figure 4 support this in that F08 SSM/I appears to be significantly better calibrated than SMMR. This should be stated in the conclusions.

---

## Author Response (AR2)

[revised manuscript text omitted]

The thorough reading of our paper by the reviewer highlighted some possible improvements that could be made to improve readers understanding. We are grateful for these comments that helped us finalize this paper.

**Response to the review**

I initially had some serious reservations regarding the authors attempt to intercalibrate SMMR Tbs using data from GMI almost 30 years later. After reading the manuscript, however, I think that the authors are providing some valuable insight into the SMMR data, although I would like to see them strengthen their conclusions a bit regarding the limitations of the SMMR data. I tend to agree that the SMMR data has value for certain applications, but any attempt to use this data to look at climate trends or more subtle variations is definitely not warranted. I believe the authors conclusions bear this out, but I would like to see a bit more explicit statement to this effect.
**Response** Indeed the proposed correction can only be used if all the hypothesis made are understood by the user. The exploration of the SMMR data teaches valuable lessons in itself regarding the instrument performances and expected accuracy from any information derived from the brightness temperatures. These limitations are made more explicit in the paper conclusion.

**Specific Comments:**

Page 3, line 17: The statement "Here, we suggest to use GMI as a reference instrument, assuming that the environmental conditions have not changed drastically from the SMMR to the GMI era, to allow the comparison of a large set of observations averaged over time. This strategy does not allow to perform a detailed intercalibration but it makes it possible to correct for major biases that so far hamper the use of SMMR over land for the generation of climate record of geophysical parameters." This statement regarding the nature of the intercalibration needs to be reiterated in the conclusions. I'm also a bit concerned with the statement suggesting that the intercalibrated data can subsequently be used for climate data records. While it depends greatly on the specific application, certainly using SMMR data to extend the global satellite time series for identifying subtle climate trends is not warranted.
**R** The paper uses data that are considered CDR, however we highlight the fact that over land the inter-calibration already performed might not be sufficient. Our proposed method is a way to improve the land surface brightness temperatures measurements for further application. Indeed it is most likely not enough to perform reliable climate analysis using this dataset. These limitations are again made clearer in the conclusion.

Page 5, Table 2: GMI has operated from March 2019 forward. The data have been reprocessed back to this date to provide a consistent data record.

**R** The data available for the GMI instrument covers the 2014-2020 period from which we have extracted the data for the years 2015 to 2017 for our analysis.

Page 5, line 13. Again, this hypothesis of limited changes in environmental conditions comes with significant caveats given climate change.
**R** This is one of the hypothesis required for our method to work. Given the existing uncertainty surface temperature changes in the past 40 years, as well as the instruments limitations we deemed it acceptable. This limitation is now also repeated in the conclusion.

Page 6, line 4: This statement is a bit unclear. Do the authors mean GMI data are collected for 2 months in summer (i.e. July and August) and two months in winter (i.e. January and February)?

**R** Exactly, to increase the number of points available for the comparison and improve the diurnal cycle constructed (taking into account the fact that SMMR only operates every other day) data from 4 months was used: January, February, July, and August from 2015 to 2017. This was made clearer in the paper.

Page 6, line 12: Did the authors attempt to look at potential calibration differences between the three years of SMMR data used (i.e. 1981, 1982 and 1987)? Is there any indication of a time-dependent change or calibration drift (or lack thereof)?

**R** This is a question that we raised also when conducting the study, however we could not notice any obvious change between the different years. A time dependent drift is still possible, but it would need access to the raw SMMR data.

Page 7, figure 1: I'm a bit unclear on the location and extent of the regions chosen. The text mentions 25 grid cells. How large is the region covered by 25 grid cells. A figure showing the various regions selected for the results in Figure 1 would be useful.

**R** To help clarify the location of the selected region the following figure was created. Each box covers roughly a 1° x 1° area. However to limit the number of figures, it was not added to the paper as the selected regions mostly serve an illustrative purpose.

[Figure]

Page 8, line 1: The relative consistency in the slopes shown in Figure 3 does lead point to the possibility of an error in either the warm load temperature or the antenna pattern correction (i.e. spillover). This might be worth mentioning in the conclusion, although the results by Dai and Che don't appear as consistent between channels.

**R** The possible sources of error are mentioned p.8, l.1 and were repeated in the conclusion. We haven't looked into the results of Dai&Che to explain these differences.

Page 8, Figure 2: The large difference between the histograms shown over Antartica (i.e. 5-10K for all except the 37 GHz V-Pol channel) raises some questions. Is this result consistent with the subsequent results shown in Figure 3? The authors mention screening out ocean and sea ice, but isn't the land region covered by snow/ice? What impact might changes in snow/ice emissivity have on the observed Tb? Emissivity

of snow/ice is difficult to determine and could potentially be influenced by a number of different factors and thus have little or nothing to do with calibration differences. As a result, unless these differences are consistent with the linear fit derived from Figure 3, it isn't clear to me that these results over Antartica are useful. Please justify.

**R** The histograms shown are in line with the results in the remaining of the paper. As a matter of fact, the same data points used for Figure 2 are also used for the estimation of the linear regression coefficient in Figure 3. The coldest data points between 200K and 240K are the ones taken over Antarctica. The issue of changing emissivity and snow coverage could have an impact on the results, as these surfaces may show more varaibility in brightness temperatures compared with other more radiative stable environments. To alleviate this issue we sampled over multiple years, mitigating the impact on the results.

Page 9, Table 3: The values of slope and intercept from the linear fit don't provide the reader a clear sense of the magnitude of the calibration differences. I suggest that the authors consider adding the mean intercalibration difference and uncertainty for GMI Tb values of 200 and 300K. This would provide the reader a much better sense of the magnitude of the adjustment as well as the uncertainly for both cold (ocean) and warm (land) scenes.

**R** The following table summarizes the expected correction at the 200K and 300K temperatures; it highlights the magnitude of the correction at the higher temperature.

| Channel | Expected correction @200K | Expected correction @300K |
| --- | --- | --- |
| 18GHz V-pol | $1.3 \pm 4.1$ | $11.3 \pm 5.2$ |
| 18GHz H-pol | $8.7 \pm 3.9$ | $13.7 \pm 4.9$ |
| 37GHz V-pol | $-2.2 \pm 4.2$ | $12.8 \pm 5.2$ |
| 37GHz H-pol | $6.8 \pm 3.9$ | $10.7 \pm 4.9$ |

Page 12, line 13: I would like to see the authors strengthen and better qualify the statement "With these hypothesis, the objective is to correct for the large differences between the sensors". As mentioned regarding Table 3, exactly how large are these differences for typical cold ocean and warm land scenes (i.e. 200 and 300 K)? Ultimately, I believe that highlighting the limitations of the SMMR data is as important if not more so than that value of the corrected data itself. The results in Figure 4 support this in that F08 SSM/I appears to be significantly better calibrated than SMMR. This should be stated in the conclusions.

**R** This remark is completely in line with the reviewer previous comments. The emphasis on the caveats of the SMMR data will be added to the conclusion and the better performance of the calibration of the later instruments will be highlighted.

The authors would like to thanks the reviewer for the comments on this paper and hope that the above responses helped clarify the questions that were raised. The work done should be useful to anyone working with SMMR data over continental surfaces.